# Ti6Al4V Alloy Remelting by Modulation Laser: Deep Penetration, High Compactness and Metallurgical Bonding with Matrix

**DOI:** 10.3390/mi13071107

**Published:** 2022-07-14

**Authors:** Longzhang Shen, Yong Chen, Hongmei Zhu, Yuantao Lei, Chanjun Qiu

**Affiliations:** 1School of Resources, Environment and Safety Engineering, University of South China, Hengyang 421001, China; shenlongzhang@usc.edu.cn; 2School of Mechanical Engineering, University of South China, Hengyang 421001, China; chenyong@usc.edu.cn (Y.C.); zhuhongmei@usc.edu.cn (H.Z.); yuantaolei@163.com (Y.L.)

**Keywords:** modulation laser, grain structure, Ti-6Al-4V, tensile strength, deep-penetration laser remelting

## Abstract

Titanium alloys are famous for their light weight, high strength, and heat- and corrosion-resistant properties. However, the excellent mechanical properties are closely related to its microstructure. Innovative machining operations are required for the welding, surface strengthening, and repairs to ensure the refining of the crystalline structure for improved strength requirements, enhanced mechanical properties, and integrating strength. By direct laser melting on the surface of Ti-6Al-4V alloy, the differences of molten pools under continuous and modulated laser mode were compared in the article. Under the same power, the heat influence zone of the laser pool could be reduced to 1/3 of that of the continuous laser. The deep molten pool could be obtained by a continuous laser by the action of high energy density. The tensile property changed a lot between different depths of melt penetration. A high-density, fine-grain molten pool could be obtained under the action of a high-frequency (20 kHz) modulation laser. The mechanical properties of the tensile sample between different depths of melt penetration, which contained the remelting zone, were close to the substrate. The research conclusions can provide technical support for the development of laser remelting processing technology.

## 1. Introduction

Titanium and its alloys are widely used in high-tech industries: aircraft and rocketry, shipbuilding, nuclear power, the chemical industry, and medicine [1]. However, titanium alloy product manufacturing is traditionally considered to be a rather difficult task [2]. The thermal history of the whole component and the microstructure after solidification play a critical role in the metallurgical and mechanical properties, which are the main concerns for engineers [3,4]. Song et al. [5] explored the effects of forming interlayers with different Selective Laser Melting (SLM) process parameters, and analyzed the defects, microstructure, and component diffusion at the interface. It was concluded that the defect of a continuous laser molten pool was difficult to avoid. At the same time, the mechanical properties of the formed parts were unstable due to the uneven distribution of the microstructure and defects. In order to obtain more uniform and anisotropic structure, Ivan et al. [1] studied the influence of the Ti6Al4V alloy track trajectories on the microstructure and mechanical properties during direct laser deposition. With the help of the competitive growth of the new grains and the potential coarsening effect during subsequent thermal-cycles, a microstructure of full equiaxed prior-β grains in Ti6Al4V fabricated by laser-directed energy deposition (DED) was finally obtained without either using any auxiliary equipment or adjusting alloy chemistry like previous research [6].

Sufficient energy density input is a necessary condition to ensure the forming of low-porosity and high-density parts [7,8]. The increase of energy density leads to the increase of temperature per unit volume, decreases the surface tension and viscosity of the molten pool, increases the width and depth of the molten pool, and makes the microstructure more compact. Moreover, the molten pool exists for a long time and the gas has enough time to escape, so the density is effectively increased. However, too high of an energy input will cause evaporation of over-burned elements, resulting in a large number of pores. It is generally believed that scanning speed plays a decisive role in laser energy input and is the direct cause of pore defects [9,10]. However, because of the difficulty of selection of technological parameters, many studies [11,12] have adopted the method of other elements’ addition to prevent the formation of defects.

Without changing the chemistry composition, another way is to apply a field-assisted laser or change the laser mode [13,14]. The field-assisted laser has its limitations, as the size of the workpiece is restricted by the volume of the auxiliary device [15]. Cha et al. [16] used pulsed laser ablation (PLA) for performing the repair of aeroengine components; in their study, follow-up fractographic and metallurgical analysis indicated that although some microstructural characteristics were different between the repair methods, the incurred surface damage was limited in magnitude to a thin surface layer (<30 μm) and the influence on fatigue life was comparable. In Bernatskyi et al. [17]’s research work, they believed that practical experience in the use of welding to solve this problem has shown the need to search technological solutions associated with increasing the depth of penetration and reducing the area of thermal effect. It was an important scientific method to search for the remelting repair method of a metal surface with a small heat-affected area and controllable structure, and it has important application value [18,19].

Laser modulation is a process in which an economical and reliable continuous fiber laser is used as a carrier. A laser has excellent temporal and spatial coherence, is similar to radio waves, easy to modulate, and the frequency of the modulation laser can be set very high. This setting can be made in order to repair cracks on the surface of the titanium alloy, and 3D printing and other modes of industry can meet the actual demand of the optical fiber laser output power and speed [20]. This paper studied the high-power, miserably temporal modulation characteristics of a glass fiber laser, which implemented the PWM-800W fiber laser with a maximum 20 kHz modulation frequency and a duty cycle adjustable modulation laser output stability. By modulating the laser to control the molten metal pool more accurately, the desired results could be obtained.

## 2. Materials and Methods

The stable modulation laser output was related to the stability and repeatability of the experimental results. Multi-path phase modulation was an electro-optic effect that can be obtained by the phase modulation of an electrical signal and the broadening of the spectrum of the output laser pulse. In the modulation process, the laser pulse output process was as follows: after the incident laser pulse was expanded and accumulated by the electro-optic crystal several times, the wide spectrum laser pulse was obtained, and the spectral laser pulse was output to achieve the purpose of the multi-path phase modulation laser pulse. The spectral characteristic of the obtained broad spectrum laser pulse was related to several parameters of multi-path phase modulation. One of the most critical parameters was the modulation signal waveform in multi-path phase modulation. The influence of the shape, width, and depth of the modulated signal waveform on the control performance of the laser pulse spectrum characteristic after phase modulation should be considered when using the modulated signal to implement multi-path phase modulation.

The continuous fiber laser produced by Kaiplin Photoelectric Technology Co., Ltd. (Tianjin, China) was used in the experiment. The model was CW1500, and the wavelength range was 1080 ± 10 nm. Phase modulation of the fiber laser used in this experiment was shown in Figure 1.

To ensure the accuracy of the results, preparatory work was necessary [21,22,23]. The national standard GB/T3621 was applied, and anything beyond standard scratches, indentation, dents, and cracks were not allowed on the surface of the sample. Sandpapers were used to polish the oxide layer on the surface of the sample until the sample surface was polished into a silver white metallic luster, and acetone was used to wipe it clean. The whole unmentioned experimentation was carried out under room temperature, which was kept at 15–30 °C, and the relative humidity did not exceed 60%. The voltage fluctuation of the laser power supply should be less than 10% to ensure the stability of the laser. In order to ensure the stability of the specimens in the cladding process, a special fixture should be used to fix the specimens.

The titanium plate used in the experiment was the Ti-6Al-4V alloy rolled plate (Guanyu Metal Material Co., Ltd., Dongguan, China), and its material composition is shown in Table 1. Low-temperature stress-relief annealing was performed at 800 °C × 1 h, AC before the experiment.

The laser process parameters, such as power, beam diameter, center gas flow, scanning speed, and so on, will have an important influence on the molten pool morphology and forming quality. It was necessary to select appropriate parameters through multiple debugs [24]. In this paper, a lot of experimental experience had been obtained before [25]: under 800 W power, using a 10 L/min central gas flow and 0.8 mm spot diameter, a better molten pool quality can be obtained. Therefore, this parameter was used in three different experiments. The difference in parameters was that the CW-800W used 800 W continuous laser output, and the scanning speed was 10 mm/s. The PWM-800 used the modulation laser obtained by the modulation mode shown in Figure 1. PWM adjustment mode was adopted, and the output was a rectangular square wave laser. The modulation pulse width was 60%. The modulation frequency was selected as 20 kHz, and the parameters of laser was listed in Table 2. By changing pulse width and modulation frequency, molten pools with various depth to width ratios can be obtained. In this paper, a 60% pulse width and 20 kHz adjustment frequency were selected to obtain a high-quality molten pool.

As shown in Figure 2, the titanium plate was remelted in the center by the laser. Wire cutting was used to cut a very thin notch (0.18 mm) on the surface of the sample. Among them, metallographic observation samples were cut out in a cross-section. The use of 200# water sanding paper thoroughly polishes off wire cut marks. Additionally, by using 400#, 600#, 800#, 1000#, 1200#, and 1500# metallographic sandpaper, the scratches on its surface were gradually covered until the surface was mirrored to prevent surface sandpaper traces from leading to misjudged tissue observation. Using a flannelette and water, the surface sandpaper traces were completely removed on a 4000 r/min metallographic polishing machine.

The etchant was configured with HF:HNO_3_:H_2_O = 1:4:10. After etching for 10–15 s, it was rinsed with water, and the surface was immediately dried thoroughly with the cold wind stop of a hair dryer [26,27]. If the sample cannot be observed immediately, it should be packed in a vacuum bag for storage. A stereomic microscope (JSZ6D, Jiangnanyongxin Co., Ltd., Nanjing, China) was used to square and measure the overall morphology of the molten pool, with a field multiple of 0.8×. A binocular inverted metallography microscope (MR2100, Jiangnanyongxin Co., Ltd., Nanjing, China) was used to observe the molten pool structure in 100× and 400× fields of view. The field scanning electron microscope (Quanta 250, FEI, Hillsboro, OR, USA) was used to observe the microstructure. The electro-hydraulic servo fatigue testing machine (PWS-100, Shidaishijin Co., Ltd., Jinan, China) was used to test the tensile strength of the sample. The size of the tensile specimen process was made according to standard GB/T228-2002A. The test force was increased with the parameter of 0.01 kN/s until the tensile strength was broken.

## 3. Results and Discussion

Single-channel laser remelting was carried out on a Ti-6Al-4V alloy surface without adding any material. There were obvious differences in the molten pool morphology between the continuous and modulated output modes. Figure 3a shows the section topography of the molten pool after remelting on the surface of the Ti-6Al-4V alloy by an 800 W continuous laser at the scanning speed of 10 mm/s. In order to better compare the changes of molten pool morphology under the three parameters, the molten pool area was determined by ImageJ software and the ratio of depth to width is listed in Table 3. In CW-800W, the molten pool area was 20.02 mm^2^. However, a modulated laser with the same 800 W power and 10 mm/s scanning speed was used, and the duty cycle was set to 60%. The modulation frequency was 20 kHz. As can be seen from Figure 3b, the molten pool has been effectively contracted with an area of only 14.5 mm^2^. However, the penetration depth did not decrease but increased slightly. It can be seen from the top of the molten pool that after continuous laser remelting, there was a detailed protrusion at the top.

According to Figure 3, the color of the molten pool in Figure 3a was darker under the same process. This is due to a large number of flaws and pores in the structure. Compared with Figure 3a, the molten pools in Figure 3b,c were significantly brighter. There was no material added in the laser remelting process, and there were varying degrees of bulges in Figure 3a–c. However, under a high-density laser, some of the material was vaporized. In general, the volume would decrease. There are two factors that are responsible for an increase in volume. One is the volume increase caused by the inclusion of defects such as pores. This will lead to a significant decrease in its mechanical properties. The second is volume expansion caused by martensitic transformation. The net gain is two-fold. Firstly, it alleviates the influence of residual stress caused by the shrinkage of the molten pool, which is dependent on rapid laser heating and cooling. Secondly, the refined grain structure also improves the microstructure strength. The volume expansion caused by martensitic transformation is in a small range. Additionally, the volume increase caused by defects depends on the size of the defect rate.

When the modulation laser scanning speed was increased to 20 mm/s, at such a small power and such a high scanning speed, the continuous laser molten pool easily showed a large area of pores and other defects [28,29]. However, under the high-frequency modulation laser of 20 kHz, the low-defect-density molten pool as shown in Figure 3b was obtained. Compared with Figure 3a,b, the heat-affected area had further shrunk. The molten pool area was as low as 7.91 mm^2^, one third of that of the continuous laser. From the angle of melting depth, a narrow and deep small melting pool can be easily obtained by the modulating laser. This technological approach was impossible to achieve in a continuous laser.

The instantaneous energy density of the pulsed laser, however, was too large, which could easily cause the evaporative removal of materials. The modulated laser has great advantages in repairing and improving surface quality because of its precise control of the molten pool morphology. By changing the laser processing parameters, both the large cone angle deep melting pool as shown in Figure 3b and the small cone angle deep melting pool as shown in Figure 4c can be obtained.

Figure 4 shows the photos of various multiples of the remelting zone structure obtained under the process parameters of the tab PWM-800W-H. Figure 4a shows the 100× field of view of the bottom remelting zone and the base material, clearly comparing the differences between the two structures. In Figure 4b,c, the two tissues shown in Figure 4a were observed in 400× magnification, respectively, to obtain high-resolution photos of the two tissues. Finally, the selection area in Figure 5c was observed under SEM to further calibrate the distribution of the structure [30].

In the process of titanium alloy remelting, it was easy to cause the uneven distribution of alloying elements, resulting in the dilution or enrichment of alloying elements in local areas. The phase transition point of the alloy in this area deviates from that of the normal alloy, resulting in various defects. According to its classification, reasons for the formation can be divided into the gap elements (oxygen, nitrogen, carbon segregation, and the segregation of alloying elements such as aluminum, molybdenum, zirconium, shown in Figure 4c remelting zone microstructure pictures). However, from the micron to tens of microns level, even in micro levels of the organization, all kinds of scale in the overall trend was obvious to the sex and homogeneity of the classification. There were no defects such as “bright strip”, “bright band”, “bright block”, “dark strip”, and “dark band” in the common macroscopic morphology of the titanium alloy [31,32]. Combined with Figure 4c,d, it can be seen that in the recrystallization process of titanium alloy, the fine equiaxed microstructure was densely covered in the lamellar α gap, and the microstructure was stirred very evenly under the action of UHF laser oscillation.

In order to compare the mechanical properties of the remelting tissues of the modulated laser, tensile tests were conducted on the substrate, continuous laser remelting sample, and modulated laser remelting sample, respectively. The sampling locations were shown in Figure 2. The tensile test results were shown in Figure 5.

The fracture strength of the base material was 944.4 MPa, and the elongation reached 13.2%. The test results show that TI-6AL-4V alloy was of high quality. After 800 W continuous laser remelting, the mechanical properties of the surface layer and bottom layer differ greatly. The surface fracture strength even exceeds the base material, reaching 955.6 MPa. Although it was only 1% stronger than the base material, this was still within the error range of the tensile test machine. However, the analysis suggests that the continuous laser of 800 W not only remelted the base material, but also carried out a heat treatment for the region where the sample was located due to its large heat-affected area, thus giving it a slightly enhanced substrate strength. However, the elongation and yield strength decreased significantly. The elongation was 9.3%, lower than the metallurgical standard of TI-6AL-4V. This was caused by the hardening of the material surface caused by the rapid cooling of the laser molten pool. In a comparison of the modulated laser remelting samples under two different scanning speeds, at double the speed, the pool heat accumulation was smaller. The heat-affected zone was also further reduced. Therefore, the mechanical properties of high-speed scanning of the modulated laser remelting sample were closest to the substrate.

In order to further determine the remelting zone structure and mechanical properties, the tensile sample was cut longitudinally as shown in Figure 2. The sample was located in the middle of the remelting zone. Half of the sample was base material and half was remelted zone. The average tensile strength was 1036 MPa and the elongation was 13.6%. Its fracture was shown in Figure 6.

Figure 6a clearly shows the difference between the remelting zone and the substrate. Figure 6b magnifies the transition zone between the two. It can be found that the dimple of fracture surface was finer than that of the substrate under modulated laser remelting. Additionally, there was a certain gradient of the dimples, which were gradually refined from the outside in when in a small position, too close to the substrate, until the gradient gradually increased. Figure 6c is a further magnification of the fracture of the substrate. The typical TI-6AL-4V fracture was presented. Dimples were a few microns in size. In Figure 6d, dimples are evenly destroyed. It shows that the grain was very uniform under the action of the UHF modulation laser. Dimples were a few microns or even submicrons in size. This shows that the remelting microstructure of the modulated laser was not only uniform, but also the grain was refined to the submicron level.

According to the experimental results, the reasons for the above phenomena were analyzed. Ti-6Al-4V alloy was mainly composed of the branch crystal with a small amount of cellular crystal. Under the same laser power, a continuous output made the shape layer of the dendrite thicker, and the pulse mode was determined by the characteristics of the two output modes, forming at the same time. The continuous output mode of heat input was relatively larger, appearing in the process of forming the two different states. The surface was quickly cooled by thermal convection, while the bottom was slowly cooled by heating the whole workpiece. The heat accumulation effect of the modulated laser was obviously improved compared with that of the continuous laser. Dendrites can grow along the vertical temperature gradient. However, under the action of the UHF laser, dendrites were rapidly formed, but were at the same time broken by the oscillating molten pool, resulting in no millimeter grain formation. Under the smaller heat input and the formation of a heat-affected zone, it was easy to form a finer crystal shape, which was beneficial to improve the mechanical properties of the forming layer.

## 4. Conclusions

A stable modulation laser output with adjustable frequency and pulse width was realized by modifying the mature continuous fiber laser technology. Additionally, the modulation frequency could be as high as 20 kHz. Ti-6Al-4V alloy was remelted by a modulation laser at different scanning speeds. Remelting technology with precise and controllable molten pool morphology could be obtained.

The microstructure formed by modulated laser remelting was analyzed. It was found that under the action of a high-frequency laser, the grain size was also refined from micro-order to nano-order. The dimples in the remelting fracture morphology were shown to be a few microns or even nanometers in size. The multiscale fine-grained structure showed good mechanical properties. These research conclusions can provide a technical support for the development of laser remelting processing technology.

## Figures and Tables

**Figure 1 micromachines-13-01107-f001:**
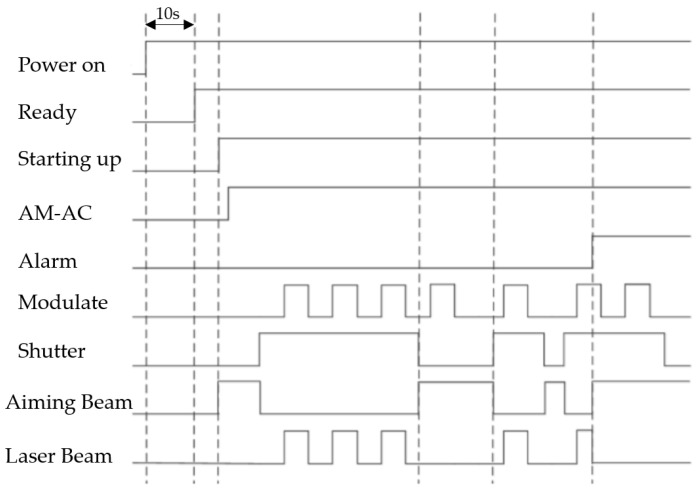
Phase modulation sequence diagram of the fiber laser.

**Figure 2 micromachines-13-01107-f002:**
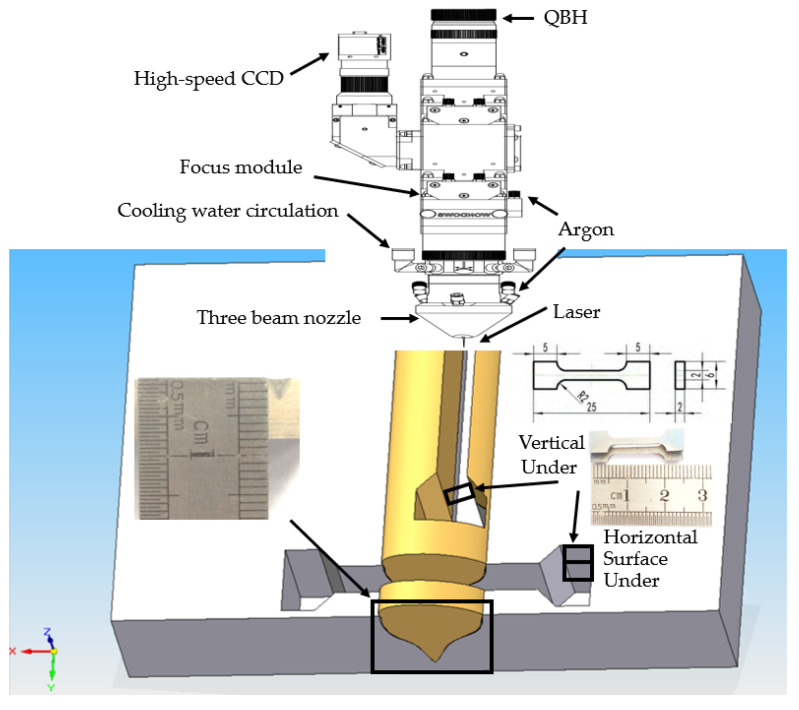
Test sampling locations.

**Figure 3 micromachines-13-01107-f003:**
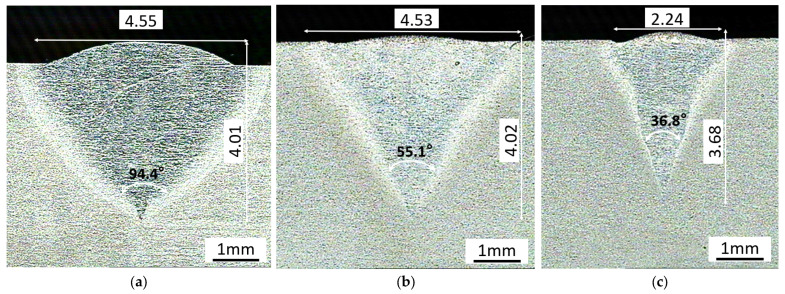
The molten pool figure of normal laser melting and deep penetration: (**a**) CW-800W (**b**) PWM-800W-L, (**c**) PWM-800W-H.

**Figure 4 micromachines-13-01107-f004:**
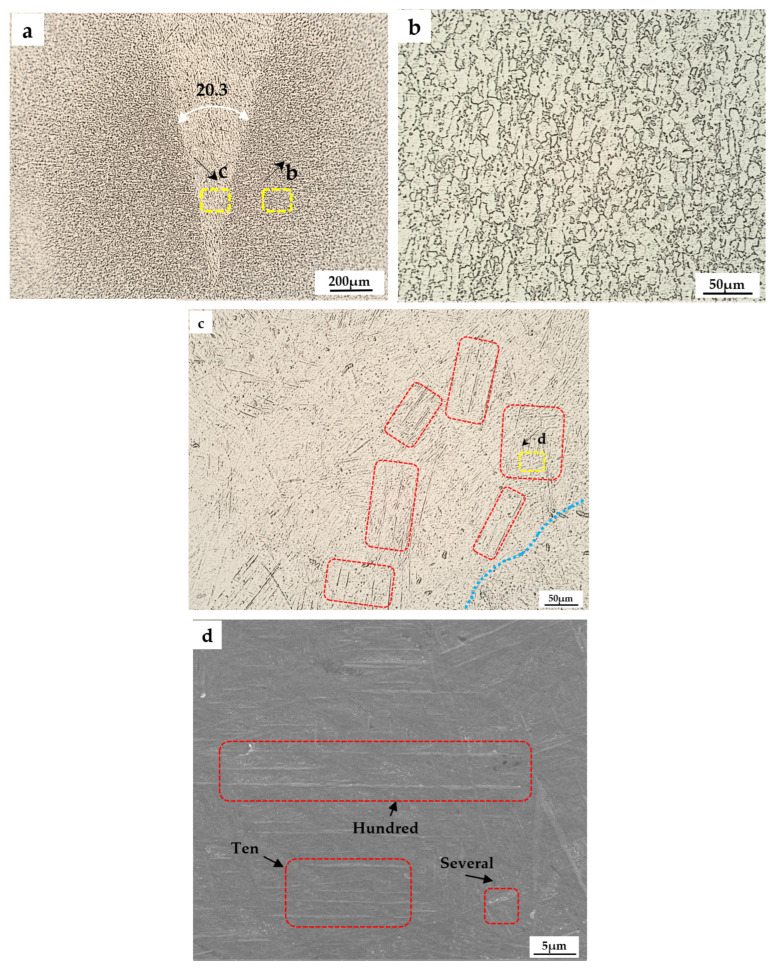
Ti-6Al-4V remelted zone grain morphologies under different magnifications: (**a**) Low magnification metallographic photo; (**b**) Enlargement in the corresponding yellow box of (**a**); (**c**) Enlargement in the corresponding yellow box of (**a**); (**d**) Enlargement in the corresponding yellow box of (**c**). Red box: typical regions.

**Figure 5 micromachines-13-01107-f005:**
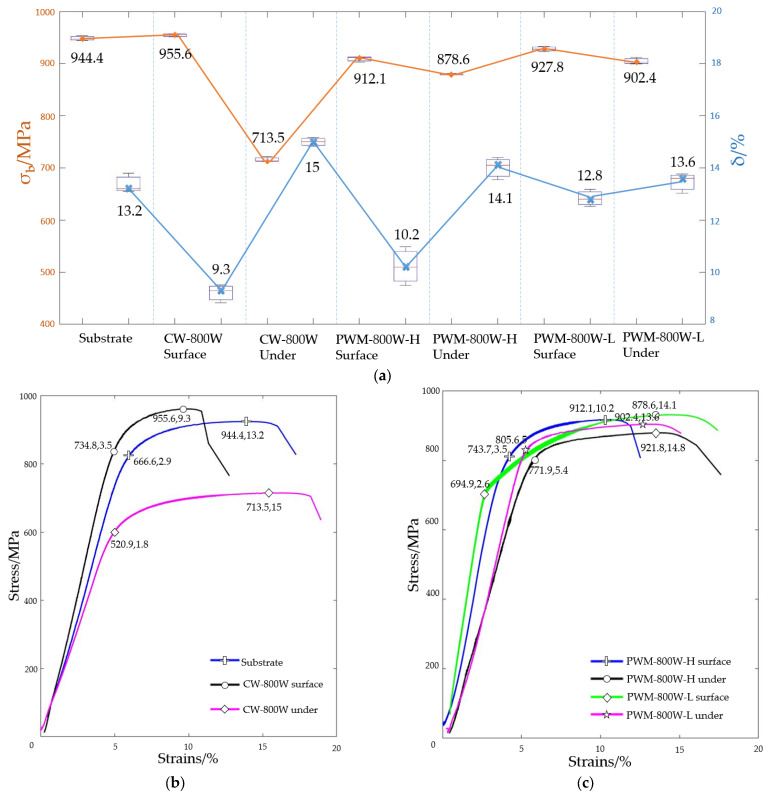
Tensile properties: (**a**) Box drawing of strength and elongation under different processes; (**b**) stress-strain curves of substrate, CW-800W surface, and under CW-800W surface; (**c**) stress-strain curves of the PWM-800-H surface, under the PWM-800-H surface, the PWM-800-L surface, and under PWM-800-L.

**Figure 6 micromachines-13-01107-f006:**
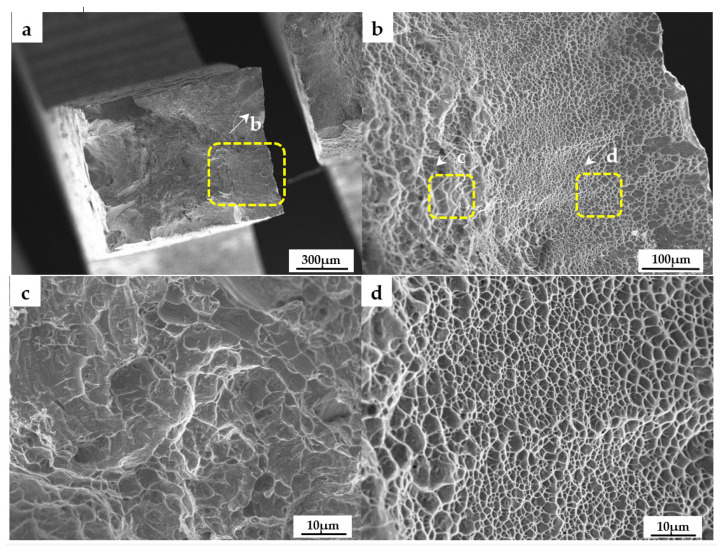
Fracture morphology under different magnification: (**a**) Overall fracture surface morphology; (**b**) Enlargement in the corresponding yellow box of (**a**); (**c**) Enlargement in the corresponding yellow box of (**b**); (**d**) Enlargement in the corresponding yellow box of (**b**).

**Table 1 micromachines-13-01107-t001:** The composition of TI-6AL-4V (wt.%).

Material	Al	V	Fe	Si	C	O	N	H	Ti
TI-6AL-4V	5.5–6.8	3.5–4.5	0.3	0.15	0.1	0.2	0.05	0.015	Balanced

**Table 2 micromachines-13-01107-t002:** Parameters of laser.

Tab	Gas Flow	Beam Diameter	Modulated Frequency	Scanning Speed
CW-800W	10 L/min	0.8 mm	-	10 mm/s
PWM-800W-L	10 L/min	0.8 mm	20 kHz	10 mm/s
PWM-800W-H	10 L/min	0.8 mm	20 kHz	20 mm/s

**Table 3 micromachines-13-01107-t003:** Parameters of the molten pool.

Tab	Pool Area/mm^2^	Penetration Angle/°	Depth-to-Width Ratio/%
CW-800W	20.02	94.4	4.01/4.55 = 0.88
PWM-800W-L	14.5	55.1	4.02/4.53 = 0.89
PWM-800W-H	7.91	36.8	3.68/2.24 = 1.64

## Data Availability

All data generated or analyzed during this study are included in this article.

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
