# Peer review of "Ti6Al4V Alloy Remelting by Modulation Laser: Deep Penetration, High Compactness and Metallurgical Bonding with Matrix"

_micromachines, 2022, doi:10.3390/mi13071107_

Round 1
Reviewer 1 Report
The language and grammar in the manuscript are very poor with a high number of typos. The manuscript requires extensive revision to improve its technical English.
Many statements are presented without accompanying backup context and thus the reader lacks context to understand the presented information. I have included an annotated manuscript with some comments and suggested changes.
I do not recommend this for publication in its current form and suggest major revisions from the authors before proceeding further.

Author Response
Firstly, thanks you very much for your in-depth and careful work on this paper. All the authors express their very deep gratitude to you.
In order to clearly express the point of this article, we modified the title of the article to "Ti6Al4V Alloy Remelting by Modulation Laser: Deep Penetration, High Compactness and Metallurgical Bonding with Matrix". And we rewrote the abstract and Introduction, focused on the explanation of the structural characteristics of the molten pool by modulation laser.
And a lot of changes had been made to the language.
Hope this is clear and we are happy to discuss this further if necessary.
Thank you again from the bottom of our heart.
Reviewer 2 Report
The manuscript under review investigates the effect of modulated laser remelting. The English language and style require extensive editing. The motivation is not clear. The literature review is not adequate and novelty is not highlighted. The tensile test data does not contain the stress-strain curve. The microstructural analysis does not bring out significant outcomes. Overall, the manuscript cannot be accepted in its present form.
Author Response
Firstly, thanks you very much for your in-depth and careful work on this paper. All the authors express their very deep gratitude to you.
In order to clearly express the point of this article, we modified the title of the article to "Ti6Al4V Alloy Remelting by Modulation Laser: Deep Penetration, High Compactness and Metallurgical Bonding with Matrix". According to your advice, the stress-strain curves were added in the section 3. And we rewrote the abstract and Introduction, focused on the explanation of the structural characteristics of the molten pool by modulation laser.
And a lot of changes had been made to the language.
Hope this is clear and we are happy to discuss this further if necessary.
Thank you again from the bottom of our heart.
Reviewer 3 Report
The article is devoted to the study of the recently very popular Ti-6Al-4V alloy. The paper requires revision and clarification of the information provided. Below are my questions and comments, which I hope will help to present the research results better.
1. The title of the article, in my opinion, does not reflect the essence of the work. When reading the introduction, one gets the impression that the article is devoted to the study of the features of the restoration of products from the Ti-6Al-4V alloy by the method of modulation laser remelting.
2. Abstract does not reflect the main and important points of the article. I suggest the authors rewrite Abstract.
3. In articles published by MDPI, Section 2 is recommended to be called Materials and Methods. «Micromachines» adheres to the same rules. I recommend the authors of the article not to deviate from the recommendations and name section 2. Materials and methods.
4. It is not clear from the paper whether some kind of surfacing material was used? Was this material in the form of wire or Ti-6Al-4V alloy powder? If used, then you need to indicate this, its chemical composition. If not, then I ask the authors to explain the photo in Figure 3. Why did the volume of metal increase? In Figure 3a, the increase in metal volume is significant.
5. The article uses two designations of the investigated alloy Ti-6Al-4V and TC4. I understand that this is the same thing, but the article needs to use one designation or give explanations to those readers who are not so familiar with these materials.
6. It is not clear from the drawing and description whether a defect was made in the titanium plate under study. In the introduction, you focused the reader's attention on fixing defects in the way you are researching. Almost nothing is said about changing the structure and properties in the introduction. At the same time, the article is mainly devoted to the study of the microstructure and mechanical properties of Ti-6Al-4V alloy after laser exposure. The introduction of the article needs to be expanded and show works devoted to the study of microstructure and mechanical properties during laser melting, here is one of the examples https://doi.org/10.3390/ma12193269
7. The Energy-dispersive X-ray spectroscopy (EDS) mapping method (https://doi.org/10.3390/machines8040079) can be used to study the chemical heterogeneity that you are talking about in lines 189-200. I think the electron microscope you have is capable of making such measurements.
Author Response
Firstly, thanks you very much for your in-depth and careful work on this paper. All the authors express their very deep gratitude to you. In response to your comments, we re-examined this article and made extensive revisions.
1. Modify the title of the article to "Ti6Al4V Alloy Remelting by Modulation Laser: Deep Penetration, High Compactness and Metallurgical Bonding with Matrix".
2.Rewrote the abstract as,
Abstract: Titanium alloys are famous for their light weight, high strength, and heat and corrosion resistant properties. However, the excellent mechanical properties are closely related to its micro-structure. Innovative machining operations are required for the weld, surface strengthening and repaired, to ensure the refining of crystalline structure for improved strength requirements, en-hanced mechanical properties and integrating strength. By direct laser melting on the surface of Ti-6Al-4V alloy, the differences of molten pools under continuous and modulated laser mode were compared in the article. Under the same power, the heat influence zone of the laser pool could be reduced to 1/3 of that of the continuous laser. The deep molten pool could be obtained by continuous laser by the action of high energy density. Whereas the tensile property changed a lot between different depth of melt penetration. High-density fine-grain molten pool could be obtained under the action of high frequency (20kHz) modulation laser. The mechanical properties of the tensile sample between different depth of melt penetration which containing the remelting zone were close to substrate. The research conclusions can provide a technical support for the development of laser remelting processing technology.
3.Title of section 2 was revised to "Materials and Methods".
4.No material added in the course of the experiment. And the reason of the volume of metal increase was explained in the second paragraph of section3.
5.Deleted "TC4" and replaced with "Ti6Al4V".
6.Rewrote the Introduction, focused on the microstructure and mechanical properties during laser melting.
7.The chemical heterogeneity of Ti6Al4V is a more complex problem, we are more likely to explain the structural characteristics of the molten pool by modulation laser.
And a lot of changes had been made to the language.
Hope this is clear and we are happy to discuss this further if necessary.
Thank you again from the bottom of our heart.
Round 2
Reviewer 2 Report
The manuscript can be accepted for publication
Reviewer 3 Report
The article has been substantially revised. The authors corrected all the shortcomings and comments. I recommend the article for publication in this version.